# Usefulness of the MALDI-TOF MS technology with membrane filter protocol for the rapid identification of microorganisms in perioperative drainage fluids of hepatobiliary pancreatic surgery

Kazuyuki Sogawa[1], Shigetsugu Takano[2]*, Takayuki Ishige[3], Hideyuki Yoshitomi[2], Shingo Kagawa[2], Katsunori Furukawa[2], Tsukasa Takayashiki[2], Satoshi Kuboki[2], Fumio Nomura[4], Masayuki Ohtsuka[2]

1 Department of Biochemistry, School of Life and Environmental Science, Azabu University, Kanagawa, Japan, 2 Department of General Surgery, Graduate School of Medicine, Chiba University, Chiba, Japan, 3 Division of Laboratory Medicine, Chiba University Hospital, Chiba, Japan, 4 Divisions of Clinical Mass Spectrometry and Clinical Genetics, Chiba University Hospital, Chiba, Japan

* stakano@faculty.chiba-u.jp

## Abstract

Surgical site infections (SSIs) are significant and frequent perioperative complications, occurring due to the contamination of the surgical site. The late detection of SSIs, especially organ/space SSIs which are the more difficult to treat, often leads to severe complications. An effective method that can identify bacteria with a high accuracy, leading to the early detection of organ/space SSIs, is needed. Ninety-eight drainage fluid samples obtained from 22 patients with hepatobiliary pancreatic disease were analyzed to identify microorganisms using matrix-assisted laser desorption/ionization-time of flight mass spectrometry (MALDI-TOF MS) with a new membrane filtration protocol and rapid BACpro® pretreatment compared to sole rapid BACpro® pretreatment. The levels of detail of rapid BACpro® pretreatment with or without filtration were also evaluated for the accuracy of bacterial identification. We found that reliable scores for *E. coli* and *E. faecalis* were obtained by inoculation with $1.0 \times 10^4$ CFU/ml after preparation of the membrane filter with rapid BACpro®, indicating approximately 10-folds more sensitive compared to sole rapid BACpro® pretreatment in drainage fluid specimens. Among 60 bacterial positive colonies in drainage fluid specimens, the MALDI-TOF MS and the membrane filtration with rapid BACpro® identified 53 isolates (88.3%) with a significantly higher accuracy, compared to 25 isolates in the rapid BACpro® pretreatment group (41.7%) ($p < 0.001$). Among the 78 strains, 14 enteric Gram-negative bacteria (93.0%) and 55 Gram-positive cocci (87.3%) were correctly identified by the membrane filtration with rapid BACpro® with a high reliability. This novel protocol could identify bacterial species within 30 min, at $2-$3 per sample, thus leading to cost and time savings. MALDI-TOF MS with membrane filter and rapid BACpro® is a quick and reliable method for bacterial identification in drainage fluids. The shortened analysis time will enable earlier selection of suitable antibiotics for treatment of organ/space SSIs to improve patients' outcomes.

**Data Availability Statement:** All relevant data are within the manuscript and the Supporting Information file.

**Funding:** This work was supported by a Grant-in-Aid for Scientific Research (KAKENHI): KIBAN C (grant no. 19K07947 to KS, grant no. 19K09113 to MO, ST, SaK), and KIBAN B (grant no. 19H03725 to ST, ShK, and MO). The URL of Grant-in-Aid for Scientific Research (KAKENHI) is as follows; https://www.jsps.go.jp/english/e-grants/ The funders had no role in study design, data collection and analysis, decision to publish, or preparation of the manuscript.

**Competing interests:** The authors have declared that no competing interests exist.

## Introduction

Surgical site infections (SSIs) are some of the most significant and frequent post-surgery complications, occurring due to the contamination of the surgical site. The mechanism underlying postoperative intra-abdominal infection remains unclear; however, bacterial contamination of the surgical site or drainage fluids is thought to promote this postoperative complication [1,2]. The late detection of SSIs, especially organ/space SSIs often leads to severe complications. Thus, even in the absence of clinical symptoms of infection, systematic cultures of drainage fluids are generally performed for the early detection of organ/space SSIs. However, conventional systematic drainage fluid culture is limited by its low predictive value and time-consuming nature [3,4]. Thus, a method that can identify bacteria with high sensitivity and specificity is urgently needed for early detection of organ/space SSIs.

Matrix-assisted laser desorption/ionization-time of flight mass spectrometry (MALDI-TOF MS) is considered a powerful tool to accurately identify pathogens, including bacteria. Recent reports describe that MALDI-TOF MS has revolutionized microbiology routine practice by decreasing time consumption at different levels [5,6]. As we have previously described [7], the MALDI Biotyper system (Bruker Daltonik, GmbH, Bremen, Germany) enables rapid detection of bacteria in clinical samples by skipping cultivation on agar plates, and is currently available for blood, urine, and peritoneal, synovial, and cerebrospinal fluids [8–10]. MALDI-TOF MS requires high bacterial counts to provide reliable scores. In previous reports, the identification rate was high for *E. coli* (97.6–100.0%) but low for *E. faecalis* (60.0–66.7%) in urine specimens with $\geq 10^5$ CFU/m$L$, showing rate variation among bacterial species [11,12]. Furthermore, inoculation with at least $1.0 \times 10^5$ CFU/m$L$ was required to obtain reliable scores for *E. coli* and *E. faecalis* after preparation by the rapid BACpro® in urine specimens [13].

In this study, we aimed to assess the clinical significance and predictive values of the MALDI Biotyper system with pretreatment membrane filtration for the direct identification of bacteria in drainage fluids after hepatobiliary-pancreatic surgery. By comparing it with a conventional method for identification of microorganisms, we demonstrated that the MALDI Biotyper system with the membrane filtration protocol is more beneficial for clinical microbiology analysis, with advantages such as higher sensitivity, higher accuracy and rapid identification. Use of this combination method might lead to the reduction of severe complications causing by organ/space SSIs and improvement of patients' clinical outcomes after surgery.

## Materials and methods

### Patients and samples

A total of 98 drainage fluid samples were prospectively obtained from 22 patients who underwent curative surgical resection for hepatobiliary pancreatic disease (hepatectomy: 8 cases, biliary operation: 4 cases, and pancreatectomy: 10 cases) in the Department of General Surgery, Chiba University Hospital, Japan, from February 2012 to June 2018. The characteristics of the patients are shown in Table 1. All the samples were randomly collected from the patients' drain which is placed close by choledochojejunostomy or pancreatojejunostomy, within 2 weeks after surgery. This was a prospective, observational study, and the study protocol (ref. number 2958) was approved the ethics committee of Chiba University, Graduate School of Medicine. Written informed consent was obtained from all the participants before surgery.

### Drainage fluids and conventional identification

Bacterial identification with the conventional method was performed using a MicroScan WalkAway system (Siemens Healthcare Diagnostics, IL, USA). For the conventional culture,

**Table 1. Clinical characteristics of samples.**

| Clinical parameters | N = 22 |
|---|---|
| Age (year; mean ± SD) | 69.5 ± 10.8 |
| Gender Male/Female | 11/11 |
| Disease (Benign/Malignant) | 2/20 |
| Malignant diseases (Liver/Bile duct/Pancreas) | 3/4/13 |

1 μ*L* of well-mixed drainage fluid was inoculated and spread on Nissui Plate Sheep Blood Agar (Nissui Pharmaceutical Co., Ltd., Tokyo, Japan) using a sterile plastic disposable loop (Eiken Chemical Co., Tokyo, Japan). Plates were incubated in an aerobic atmosphere at 37˚C for 18 to 24 h. When bacterial growth was observed, the colonies on blood agar were counted, and colonies from plates were identified using the MicroScan WalkAway system.

## 16S rRNA gene sequencing and sequence analysis

The total DNA of fresh colonies was prepared using a MagNA Pure Compact DNA isolation kit I (Roche Molecular Biochemicals, Mannheim, Germany). Polymerase chain reaction (PCR) amplification of the 16S rRNA gene was performed as previously described by Okazaki *et al.* [14] and Otsuka *et al.* [15], using primers 8UA (5′-AGA GTT TGA TC(A/C) TGG CTC AG-3′) and 1485B (5′-TAC GGT TAC CTT GTT ACG AC-3′). Amplicons were purified and sequenced using primers 519A (5′-CAG C(A/C)G CCG CGG TAA T-3′), 519B (5′-ATT ACC GCG GC(G/T) GCT G-3′), 907A (5′-AAA CT(T/C) AAA (T/G)GA ATT GAC GG-3′), and 907B (5′-CCG TCA ATT C(A/G) TTT (A/G)A GTT T-3′). PCR products were sequenced via Sanger sequencing using 3730XL DNA analyzer (Applied Biosystems). A homology search of 16S rRNA gene sequences was performed against sequences registered in GenBank/EMBL/DDBJ using BLAST. Final sequencing identifications of 16S rRNA was assigned according to Clinical and Laboratory Standards Institute (CLSI) interpretive criteria.

## Membrane filtration

Ten milliliters of the drainage fluid were centrifuged at 2,000×*g* for 30 sec. The supernatant was manually drawn into a 10 m*L* syringe fitted with a 0.30 μm MF-Millipore membrane filter (Merck KGaA, Darmstadt, Germany). This procedure was completed within 2 min. Subsequently, each membrane was manually washed with 5 mL of rapid BACpro® reaction buffer 1 delivered using a 10 m*L* syringe with a 0.30 μm membrane quality monitor. This procedure was completed within 2 min.

## Rapid BACpro®

Drainage fluid sample preparation was also performed using the rapid BACpro® kit (Nittobo Medical Co., Tokyo, Japan) [16]. In brief, drainage fluid (10 m*L*) was centrifuged at 2,000×*g* for 30 sec. The supernatant was then centrifuged at 15,000×*g* for 10 min for bacterial collection. The pellet samples were sequentially combined with 10 μ*L* of reaction buffer 1, 100 μ*L* of reaction buffer 2, and 10 μ*L* of polymer suspension. The mixture was separated using a desktop centrifuge (Chibitan R, Merck Millipore, Billerica, MA, USA), at 2,000×*g* for 30 sec, and the supernatant was discarded. Then, the resulting aggregate was resuspended in 1,000 μ*L* of 70% acetonitrile, and the supernatant was separated by centrifugation at 2,000×*g* for 30 sec. Finally,

the obtained precipitate was resuspended in 30 μL of 70% formic acid and 100 μL of 100% ace-tonitrile, and the supernatant was separated by centrifugation at 2,000×*g* for 60 sec.

## Bacterial identification by MALDI-TOF MS

The MALDI-TOF α-cyano-4-hydroxycinnamic acid matrix was prepared daily as a saturated solution in 50% acetonitrile and 2.5% trifluoroacetic acid. Subsequently, 1 μL of the sample extract (prepared using either of the two approaches above) was spotted on a steel target plate (Bruker Daltonik) and allowed to dry. Next, 1 μL of matrix solution was added and air dried. The target plate was then placed in the MALDI-TOF MS apparatus. To identify the isolates, MALDI-TOF MS was performed on an AutoFlex® TOF/TOF mass spectrometer equipped with Flexcontrol™ software v. 3.0 (Bruker Daltonik) for automatic acquisition of mass spectra in the linear positive mode within a range of 2 to 20 kDa. Laboratory technicians performed all MALDI-TOF MS measurements in the study.

The Autoflex® II TOF/TOF mass spectrometer was periodically calibrated using the Bruker Daltonik bacterial test standard (*Escherichia coli* extracts containing RNase A and myoglobin). Automated analysis of raw spectral data was performed using MALDI Biotyper automation v.3.0 software (Bruker Daltonik) with a library of 5,989 spectra (database updated on July 31, 2015) and default settings. The whole process, from MALDI-TOF MS measurement to identi-fication, was performed automatically without user intervention. After alignment, peaks with a mass-to-charge ratio difference of <250 ppm were deemed identical. The peak lists generated were matched against the reference library by using an integrated pattern matching algorithm in the software. Pattern-matching results are expressed as scores ranging from 0 to 3, with a score <1.7 not considered to give reliable identification and a score ≥2.0 indicating identifica-tion of a species [7]. The Sepsityper module software (Bruker Daltonik) was utilized for cul-tures with two different bacterial species [17].

## Limitation of detection range for MALDI-TOF MS

In order to determine the minimal bacterial concentration allowing reliable MALDI-TOF MS identifications, we inoculated 50 m*L* aliquots of sterile water with one strain of *E. coli* or *Enterococcus faecalis* (*E. faecalis*) at a bacterial count of $1.0 \times 10^8$ colony-forming unit (CFU)/m*L*. We performed sequential dilutions in order to achieve aliquots of each microorganism at the following bacterial counts: $1.0 \times 10^6$, $5.0 \times 10^5$, $1.0 \times 10^5$, $5.0 \times 10^4$, $1.0 \times 10^4$, $5.0 \times 10^3$ and $1.0 \times 10^3$ CFU/m*L*. When necessary, aliquots were further diluted before spreading on a plate, in order to obtain a countable number of colonies (50 to 500). Three 100 μL aliquots of each dilution were plated onto blood agar plates. These plates were incubated at 37˚C in an aerobic atmosphere, and colonies were counted manually. The mean of three aliquots of each dilution was defined as the final count for the aliquot. Ten milliliter samples of each dilution were taken for the preparation kit for MALDI-TOF MS, which was performed according the method previously described.

## Results

### Increase in MALDI-TOF MS scores by the membrane filtration with BACpro®

In this study, we found that reliable scores for *E. coli* and *E. faecalis* were obtained by inocula-tion with $1.0 \times 10^4$ CFU/m*l* after preparation of the membrane filter with rapid BACpro® in drainage fluid specimens (Table 2). While an inoculation of at least $1.0 \times 10^5$ CFU/mL is neces-sary for existing measurement kits, BACpro® to reliably identify the bacterial population, our

**Table 2. MALDI-TOF MS scores for bacteria count of Escherichia coli and Enterococcus faecalis.**

| Pretreatment protocol | Microorganism | MALDI-TOF MS score for bacteria count (CFU/ml) | | | | | | |
|---|---|---|---|---|---|---|---|---|
| | | $1.0\times10^6$ | $5.0\times10^5$ | $1.0\times10^5$ | $5.0\times10^4$ | $1.0\times10^4$ | $5.0\times10^3$ | $1.0\times10^3$ |
| rapid BACpro® | E. coli | 2.297 | 2.188 | 2.136 | 1.994 | 1.975 | 1.741 | 1.621 |
| | E. faecalis | 2.165 | 2.082 | 2.041 | 1.741 | 1.722 | 1.684 | 1.611 |
| Membrane filter with rapid BACpro® | E. coli | 2.388 | 2.351 | 2.254 | 2.213 | 2.064 | 1.984 | 1.748 |
| | E. faecalis | 2.257 | 2.185 | 2.094 | 2.067 | 2.004 | 1.847 | 1.649 |

newly established pretreatment method using the membrane filter with rapid BACpro® can identify it from $1.0 \times 10^4$ CFU/m$L$. These results demonstrate that the membrane filter with rapid BACpro® increases the sensitivity (approximately 10-folds higher than the sensitivity of rapid BACpro® without the membrane filter) for microorganism identification in drainage fluids.

## Drainage fluid results of specimens available for MALDI-TOF MS

Of 98 drainage fluid specimens, growth of colonies was observed in 60 (61.2%) specimens: 42 with single-colony morphology, and 18 with two-colony morphology (Table 3). Thirty-eight specimens did not grow in culture, and MALDI-TOF MS did not identify a significant protein profile in any of these cases. In specimens with single-colony morphology, MALDI-TOF MS correctly identified 21 isolates (50.0%) using rapid BACpro®, whereas MALDI-TOF MS and the membrane filtration with rapid BACpro® identified 37 isolates (88.1%) with a significantly higher accuracy (p < 0.001; chi-squared test). In the 21 cases of rapid BACpro® in which identification was not possible, the bacterial count was ≤$1.0\times10^5$ CFU/m$L$, which was below the detection limit. In the five cases of filtration protocol with rapid BACpro® in which identification was not possible, the bacterial count was ≤$1.0\times10^4$ CFU/m$L$, which was below the detection limit. Consistent with this, the filtration protocol with rapid BACpro® (80.0%) showed

**Table 3. Comparison of drainage fluid specimens analyzed by MALDI-TOF MS with two pretreatment methods.**

| Drainage fluid culture (no. of isolates) | rapid BACpro® (no. of isolates) | Membrane filter with rapid BACpro® (no. of isolates) |
|---|---|---|
| Growth of colonies (N = 60) | | |
| 1-colony morphology (N = 42) | Positive with same identification (21)[a] | Positive with same identification (37)[a] |
| | No reliable identification (21)[b] | No reliable identification (5)[c] |
| 2-colony morphology (N = 18) | E. faecalis and E. feacium (2) | E. aerogenes and E. faecalis (2) |
| | E. faecalis and K. pneumoniae (1) | E. faecalis and E. feacium (9) |
| | E. faecalis and P. aeruginosa (1) | E. feacium and E. raffinosus (1) |
| | No reliable identification (14)[b] | E. faecalis and K. pneumoniae (1) |
| | | E. faecalis and P. aeruginosa (1) |
| | | E. feacium and S. epidermidis (1) |
| | | K. pneumoniae and S. epidermidis (1) |
| | | No reliable identification (2)[c] |
| No growth (N = 38) | | |

no.: number,

[a] Identification at the species level,

[b] Drainage fluid specimens were cultured with <$1.0\times10^5$ CFU/m$l$,

[c] Drainage fluid specimens were cultured with <$1.0\times10^4$ CFU/m$l$.

higher accuracy for bacterial identification compared to that of rapid BACpro® (20.0%) in specimens with two-colony morphology (p < 0.001; chi-squared test). In two-colony morphology samples in which both species were identified, the presence of a mixed culture was recognized by the Sepsityper module software. These results suggest that the method of filtration protocol with rapid BACpro® is superior to the rapid BACpro® method in terms of accuracy in bacterial identification.

Next, we compared the accuracy of the two different pretreatments for microorganism identification (Table 4). Seventy-eight strains were identified using the conventional method, including 15 enteric Gram-negative bacteria (GNB) and 63 Gram-positive cocci (GPC). The 78 strains identified by the conventional method were subjected to 16S rRNA analysis (S1 Table), and the identification results were identical in all the strains when the bacterial counts were more than $1.0 \times 10^4$ CFU/m$L$. MALDI-TOF MS correctly identified 7 enteric GNB (46.0%) and 23 GPC (36.5%) by the rapid BACpro, in comparison with 14 enteric GNB (93.0%) and 55 GPC (87.3%) by the membrane filtration with rapid BACpro. Overall, among the 78 specimens with colony counts $\geq 1.0 \times 10^5$ CFU/m$L$ or $\geq 1.0 \times 10^4$ CFU/m$L$, microorganism identification by the conventional method coincided with the rapid BACpro® in 30 cases (38.5%) and membrane filtration with rapid BACpro® in 69 cases (88.5%), suggesting a significant difference between two pretreatment methods (p < 0.001; chi-squared test). These results indicated that the accuracy of microorganism identification using membrane filtration with rapid BACpro® is superior to that in sole rapid BACpro® pretreatment.

Of the GNB, *Enterobacter aerogenes*, *Enterobacter cloacae*, *Klebsiella pneumoniae*, and *Pseudomonas aeruginosa* were correctly identified in 1 (20.0%), 1 (100.0%), 3 (60.0%) and 2 (50.0%) samples using the rapid BACpro®, and in 5 (100.0%), 1 (100.0%), 5 (100.0%) and 3 (75.0%) samples using the membrane filtration with rapid BACpro®, respectively. Of the GPC, *Enterococcus faecalis*, *Enterococcus feacium*, *Enterococcus raffinosus*, *Staphylococcus epidermidis*, and *Staphylococcus aureus* were correctly identified in 12 (41.4%), 10 (35.7%), 1

**Table 4. Identification of 78 microorganisms by MALDI-TOF MS with different pretreatment methods.**

| Conventional method (no. of isolates) | 16S rRNA method (no. of isolates) | rapid BACpro® (no. of isolates) | Membrane filter with rapid BACpro® (no. of isolates) |
|---|---|---|---|
| *Enterobacter aerogenes* (5) | *Enterobacter aerogenes* (5) | *Enterobacter aerogenes* (5) | *Enterobacter aerogenes* (5) |
| | | No reliable identification (4) | |
| *Enterobacter cloacae* (1) | *Enterobacter cloacae* (1) | *Enterobacter cloacae* (1) | *Enterobacter cloacae* (1) |
| *Klebsiella pneumoniae* (5) | *Klebsiella pneumoniae* (5) | *Klebsiella pneumoniae* (5) | *Klebsiella pneumoniae* (5) |
| | | No reliable identification (2) | |
| *Pseudomonas aeruginosa* (4) | *Pseudomonas aeruginosa* (4) | *Pseudomonas aeruginosa* (4) | *Pseudomonas aeruginosa* (3) |
| | | No reliable identification (2) | No reliable identification (1) |
| Total gram-negative bacteria (15) | 100.0% (15/15) | 46.0% (7/15) | 93.0% (14/15) |
| *Enterococcus faecalis* (29) | *Enterococcus faecalis* (29) | *Enterococcus faecalis* (29) | *Enterococcus faecalis* (26) |
| | | No reliable identification (17) | No reliable identification (3) |
| *Enterococcus feacium* (28) | *Enterococcus feacium* (28) | *Enterococcus feacium* (28) | *Enterococcus feacium* (25) |
| | | No reliable identification (18) | No reliable identification (3) |
| *Enterococcus raffinosus* (1) | *Enterococcus raffinosus* (1) | No reliable identification (1) | *Enterococcus raffinosus* (1) |
| *Staphylococcus epidermidis* (4) | *Staphylococcus epidermidis* (4) | *Staphylococcus epidermidis* (1) | *Staphylococcus epidermidis* (2) |
| | | No reliable identification (3) | No reliable identification (2) |
| *Staphylococcus aureus* (1) | *Staphylococcus aureus* (1) | No reliable identification (1) | *Staphylococcus aureus* (1) |
| Total gram-positive cocci (63) | 100.0% (63/63) | 36.5% (23/63) | 87.3% (55/63) |
| Total (78) | 100.0% (78/78) | 38.5% (30/78) | 88.5% (69/78) |

(100.0%), 1 (25.0%) and 0 (0.0%) samples using the rapid BACpro®, and in 26 (89.7%), 25 (89.3%), 1 (100.0%), 2 (50.0%) and 1 (100.0%) samples using the filtration protocol with rapid BACpro®, respectively. In both categories of bacteria, the protocol of membrane filter with rapid BACpro® showed higher accuracy for microorganism identification compared to the rapid BACpro® pretreatment in this study.

## Discussion

MS is a powerful and reliable tool that can comprehensively identify target molecules with high specificity. MS has been used on a global scale and can be applied to various sub-disciplines in laboratory medicine. Among all applications, the most successful application of MS is the identification of microorganisms using MLDI-TOF MS [18]. Identification of microorganisms is primarily undertaken by the culture of bacterial colonies on agar plates. The MALDI-TOF MS system has been developed for bacterial identification in bacterial colonies. For the identification of monomicrobial and polymicrobial samples using MLDI-TOF MS, the Sepsityper module which we utilized also includes a novel functionality where an alert is generated automatically for mixed samples [17]. Subsequently, it was tested on clinical specimens in clinical and diagnostic microbiology laboratories worldwide [5,18–23]. In this study, we demonstrated that MALDI-TOF MS with the filtration protocol can be used to identify microorganisms in postoperative drainage fluids with high sensitivity and accuracy. We developed our in-house membrane filtration method to ensure high sensitivity and to enhance the identification rates of microorganisms in drainage fluids.

After major surgeries such as hepato-biliary pancreatic surgery, organ/space SSIs may result in severe complications that are life threatening. Specifically, infections in the pancreatic fistula after pancreaticoduodenectomy can result in intra-abdominal bleeding, which is closely associated with increased morbidity and mortality. In this study, we focused on identifying microorganisms in drainage fluids, as early identification may allow for the early selection of suitable antibiotics to treat organ/space SSIs and thus reduce morbidity and mortality. Indeed, internal or external drainage is the first choice of treatment for fluid collection with bacterial infection in organ/space SSIs. Occasionally, in addition to the difficulties of interventional drainage for abdominal rest abscess, symptoms of organ/space SSIs, such as fever or abdominal discomfort, also emerge after drainage tube removal. Therefore, administration of suitable antibiotic agents would be one of the best treatments for organ/space SSIs in clinical settings. Sugiura et al. have described that intraoperative bacterial contamination, which is significantly associated with positive bacterial culture of abdominal drain fluid on day 1 and day 3 post-operation, can lead to the development of organ/space SSIs and severe pancreatic fistula as well as longer hospital stays following pancreaticoduodenectomy [24]. Therefore, accurate identification of microorganisms in drainage fluids will help provide good clinical care to patients after operations.

Conventional protocols for bacterial identification involve additional hands-on processing, which are both labor-intensive and costly. To overcome these major disadvantages, recent molecular assays such as PCR technologies have been developed to improve the turnaround time of blood and urine cultures in microbiology experiments. These approaches can significantly reduce the turnaround time and have been approved by the U.S. Food and Drug Administration. Nevertheless, their initial application usually requires 2 to 4 instruments, and the price per specimen test ranges from approximately \$15 to \$300. Additionally, target detection is limited owing to the nature of these assays [14,25,26]. An alternative approach that can overcome these limitations is the use of MALDI-TOF MS of positive blood cultures and urine. Importantly, a previous study clearly demonstrated that transition from conventional

microbiological methods to MALDI-TOF MS resulted in significant cost savings, which was estimated to be 51.7% of total costs during a 12-month study [27]. In line with this benefit, the total procedural cost for microorganism identification in drainage fluid using MALDI Bioty-per with membrane filter and rapid BACpro® is very low, at approximately $3-$5 per sample. Furthermore, the conventional (gold standard) method for identifying drainage fluid culture (i.e., quantitative culture of clinical samples on solid medium followed by biochemical characterization of isolates) requires 48 to 72 hours [28]. In contrast, MALDI-TOF MS can take shorter time of process for identification of bacteria from culture in 24 hours. Furthermore, MALDI-TOF MS of drainage fluid specimens prepared using the membrane filtration proto-col with rapid BACpro® can identify bacterial species using the MALDI Biotyper automation software in approximately 30 min, and PCR and microarrays can be completed within approximately 1 hour [14,25,26]. Therefore, this method may be an attractive option for bacterial identification in terms of time and cost savings.

There are several limitations to this study. One such limitation is that this is a single-center study, with results obtained from a specific management protocol. Secondary, the presence of anaerobes has not been investigated. In addition, although MALDI-TOF MS is now an essential tool for rapid pathogenic bacterial identification, antimicrobial susceptibility testing of causative agents is another important task in microbiology. Currently, investigations into bacterial identification, such as detection of antibiotic modification and antimicrobial resistance based on identified peaks in mass spectra as well as the inhibition of bacterial growth using existing antibiotic agent, have been conducted in microbiology [29]. Although the application of MALDI-TOF MS for detecting bacterial resistance to antibiotics has been recently demonstrated, its utility is still being explored [30–35]. Further studies are required to establish a method for assessing antimicrobial susceptibility to causative agents using MALDI-TOF MS. In conclusion, MALDI-TOF MS with membrane filter and rapid BACpro® is a speedy and reliable protocol for identifying bacteria in drainage fluids in a clinical setting. It allows for reduced analysis time with high accuracy that helps earlier and better selection of antibiotics for the treatment of organ/space SSIs to improve patient outcomes during the perioperative period.

## Supporting information

**S1 Table. The 78 strains identified by the conventional method were subjected to 16S rRNA analysis.**
(XLSX)

## Acknowledgments

We thank the staff of Department of General Surgery for their contributions to this work.

## Author Contributions

**Conceptualization:** Kazuyuki Sogawa, Shigetsugu Takano, Hideyuki Yoshitomi.

**Data curation:** Kazuyuki Sogawa, Takayuki Ishige.

**Formal analysis:** Kazuyuki Sogawa, Shigetsugu Takano, Tsukasa Takayashiki.

**Funding acquisition:** Kazuyuki Sogawa, Shigetsugu Takano, Masayuki Ohtsuka.

**Investigation:** Kazuyuki Sogawa, Takayuki Ishige, Shingo Kagawa, Satoshi Kuboki.

**Methodology:** Kazuyuki Sogawa, Shigetsugu Takano, Hideyuki Yoshitomi, Katsunori Furukawa.

**Project administration:** Kazuyuki Sogawa, Shigetsugu Takano, Hideyuki Yoshitomi.

**Supervision:** Kazuyuki Sogawa, Shigetsugu Takano, Fumio Nomura, Masayuki Ohtsuka.

**Validation:** Kazuyuki Sogawa, Takayuki Ishige.

**Visualization:** Kazuyuki Sogawa, Shigetsugu Takano.

**Writing – original draft:** Kazuyuki Sogawa, Shigetsugu Takano, Fumio Nomura, Masayuki Ohtsuka.

**Writing – review & editing:** Kazuyuki Sogawa, Shigetsugu Takano, Takayuki Ishige, Hideyuki Yoshitomi, Shingo Kagawa, Katsunori Furukawa, Tsukasa Takayashiki, Satoshi Kuboki, Fumio Nomura, Masayuki Ohtsuka.

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
