## [Decision Letter · Decision Letter 0]

18 Nov 2020

PONE-D-20-20446

Usefulness of the MALDI-TOF MS technology with membrane filter protocol for the rapid identification of microorganisms in perioperative drainage fluids of hepatobiliary pancreatic surgery.

PLOS ONE

Dear Dr. Takano

Thank you for submitting your manuscript to PLOS ONE. After careful consideration, we feel that it has merit but does not fully meet PLOS ONE’s publication criteria as it currently stands. Therefore, we invite you to submit a revised version of the manuscript that addresses the points raised during the review process.

We look forward to receiving your revised manuscript.

Kind regards,

Massimiliano Galdiero, M.D., Ph.D.

Academic Editor

PLOS ONE

Journal Requirements:

Additional Editor Comments (if provided):

Reviewers' comments:

Reviewer's Responses to Questions

**Comments to the Author**

1. Is the manuscript technically sound, and do the data support the conclusions?

Reviewer #1: Yes

Reviewer #2: Yes

2. Has the statistical analysis been performed appropriately and rigorously? 

Reviewer #1: Yes

Reviewer #2: N/A

3. Have the authors made all data underlying the findings in their manuscript fully available?

Reviewer #1: Yes

Reviewer #2: Yes

4. Is the manuscript presented in an intelligible fashion and written in standard English?

Reviewer #1: Yes

Reviewer #2: Yes

5. Review Comments to the Author

Reviewer #1: Very interesting and innovative study, well performed, with accuracy and precision.

I have only minor comments:

- I recommend to slightly reformulate, in the abstract, the methods paragraph, in order to include a sentence about the tests made to evaluate the LOD of Rapid BACpro with/without filtration

- In the result chapter, please report the details about the identification rates also for cultures with 2 different bacterial species, and report how the MALDI-TOF MS detected both species from the direct sample (in all cases, were both species identified by MALDI? how did you realized, from the MALDI ID result, that two different species were present?)

- In the discussion chapter, please discuss the performance of MALDI-TOF MS in monomicrobial and polymicrobial samples.

- Please replace BioTyper with Biotyper, and add mention "Bruker Daltonik" as "Bruker Daltonik GmbH, Bremen, Germany)

- Please mention that one of the limitation of the study is that the presence of anaerobes has not been investigated (since the plates were incubated only in aerobic environment) - the 38 culture-negative samples could actually be positive for the presence of anaerobes.

Reviewer #2: The manuscript "Usefulness of the MALDI-TOF MS technology with membrane filter protocol for the rapid identification of microorganisms in perioperative drainage fluids of hepatobiliary pancreatic surgery" by Kazuyuki Sogawa et al. describes a novel approach to identify microbes directly from clinical specimen by MALDI-TOF mass spectrometry. The Authors modified the protocol using a novel commercial system BACpro for increasing the sensitivity of detected bacteria.

Generally, the use of clinical specimen for direct identification of bacteria is challenging. Currently used methods (e.g., commercially available SepsiTyper) are usually time-consuming with limited outputs. The Authors show another approach which could provide reliable data within short turnaround time.

The study is sufficiently designed and the manuscript is well written. In the Discussion, limits of the study are also provided.

I have some minor comments to the text only:

- Page 7, Line 5: The Authors can avoid the use of "(on average)" as the time difference is probably meaningless;

- Page 8, Lines 7-10: It is not necessary to specify MWs of proteins included in commercial calibration standard;

- Page 8, Lines 14-15: Similarly, description of automatic spectra processing can be avoided;

- Page 10, Line 16: Please use capital "R" for Gram-negative and Gram-positive;

- Page 11, Lines 19-20: Please reformulate "The gold standard for microorganism identification is to culture bacterial colonies on agar plates" as the culture is only prerequisite for identification of microbes.

- Page 11, Lines 20-21: MALDI-TOF MS was not originally developed for identification of microbes. Please reformulate.

- Page 12, Line 20: "for bacterial purification" - I am very sorry to say, but I do not understand this sentence;

- Page 12, Lines 22-24: Microarrays are not routinely used in diagnostic laboratories. If I am not correct, please add the reference;

Page 13, Lines 9-12: If MALDI-TOF MS is used for identification of bacteria from culture, the process can take shorter (24 h) time. Please correct.

6. PLOS authors have the option to publish the peer review history of their article (what does this mean?). If published, this will include your full peer review and any attached files.

Reviewer #1: **Yes: **Miriam Cordovana, Bruker Daltonik, GmbH

Reviewer #2: **Yes: **Jaroslav Hrabak

---

## [Author Response · Author response to Decision Letter 0]

3 Dec 2020

Thank the academic editor and two reviewers so much for taking your precious time to evaluate and improve our manuscripts and giving us the precious comments. 

Reviewer #1:

- I recommend to slightly reformulate, in the abstract, the methods paragraph, in order to include a sentence about the tests made to evaluate the LOD of Rapid BACpro with/without filtration.

[Authors' reply] We appreciate the reviewer’s suggestion. We added the sentence, “The levels of detail of rapid BACpro® pretreatment with or without filtration were also evaluated for the accuracy of bacterial identification.” in the section of Abstract of the revised manuscript.

- In the result chapter, please report the details about the identification rates also for cultures with 2 different bacterial species, and report how the MALDI-TOF MS detected both species from the direct sample (in all cases, were both species identified by MALDI? how did you realized, from the MALDI ID result, that two different species were present?)

[Authors' reply] We appreciate the reviewer’s meaningful suggestion. We added the sentence, “In two-colony morphology samples in which both species were identified, the presence of a mixed culture was recognized by the Sepsityper module software.” in the section of Result of the revised manuscript.

- In the discussion chapter, please discuss the performance of MALDI-TOF MS in monomicrobial and polymicrobial samples.

[Authors' reply] We appreciate the reviewer’s helpful comment. We added the sentence, “For the identification of monomicrobial and polymicrobial samples using MLDI-TOF MS, the Sepsityper module which we utilized also includes a novel functionality where an alert is generated automatically for mixed samples [17].” in the section of Discussion of the revised manuscript.

We also added the article “17. Scohy A, Noël A, Boeras A, Brassinne L, Laurent T et al. Evaluation of the Bruker MBT Sepsityper IVD module for the identification of polymicrobial blood cultures with MALDI-TOF MS. Eur J Clin Microbiol Infect Dis 2018;37:2145–2152.” in the reference of the revised manuscript.

- Please replace BioTyper with Biotyper, and add mention "Bruker Daltonik" as "Bruker Daltonik GmbH, Bremen, Germany)

[Authors' reply] As the reviewer‘s suggestion, we amended “BioTyper” to “Biotyper”, and "Bruker Daltonics" to "Bruker Daltonik" in the revised manuscript.

- Please mention that one of the limitation of the study is that the presence of anaerobes has not been investigated (since the plates were incubated only in aerobic environment) - the 38 culture-negative samples could actually be positive for the presence of anaerobes.

[Authors' reply] We appreciate the reviewer’s comment. We added the limitation indicated above in the revised manuscript.

Reviewer #2:

- Page 7, Line 5: The Authors can avoid the use of "(on average)" as the time difference is probably meaningless;

[Authors' reply] We appreciate the reviewer’s comment. We deleted the part "(on average)" in the revised manuscript.

- Page 8, Lines 7-10: It is not necessary to specify MWs of proteins included in commercial calibration standard;

[Authors' reply] As the reviewer’s helpful suggestion, we deleted the part indicated above in the revised manuscript.

- Page 8, Lines 14-15: Similarly, description of automatic spectra processing can be avoided;

[Authors' reply] As the reviewer’s helpful suggestion, we also deleted the part indicated above in the revised manuscript.

- Page 10, Line 16: Please use capital "R" for Gram-negative and Gram-positive;

[Authors' reply] We amended “gram-negative and gram-positive “ to “Gram-negative and Gram-positive” in the revised manuscript.

- Page 11, Lines 19-20: Please reformulate "The gold standard for microorganism identification is to culture bacterial colonies on agar plates" as the culture is only prerequisite for identification of microbes.

[Authors' reply] As the reviewer’s suggestion, the sentence was amended to “Identification of microorganisms is primarily undertaken by the culture of bacterial colonies on agar plates.” in the revised manuscript.

- Page 11, Lines 20-21: MALDI-TOF MS was not originally developed for identification of microbes. Please reformulate.

[Authors' reply] We agree with the comment. We reformulated the sentence “The MALDI-TOF MS system has been developed for bacterial identification in bacterial colonies.” in the revised manuscript.

- Page 12, Line 20: "for bacterial purification" - I am very sorry to say, but I do not understand this sentence;

[Authors' reply] We appreciate the comment. In order to avoid confusion, we amended “ for bacterial purification” to “ for bacterial identification” in the revised manuscript.

- Page 12, Lines 22-24: Microarrays are not routinely used in diagnostic laboratories. If I am not correct, please add the reference;

[Authors' reply] We deleted “Microarrays” in the revised manuscript.

- Page 13, Lines 9-12: If MALDI-TOF MS is used for identification of bacteria from culture, the process can take shorter (24 h) time. Please correct.

[Authors' reply] We added the sentence “MALDI-TOF MS can take shorter time of process for identification of bacteria from culture in 24 hours.” in the revised manuscript.

---

## [Decision Letter · Decision Letter 1]

12 Jan 2021

Usefulness of the MALDI-TOF MS technology with membrane filter protocol for the rapid identification of microorganisms in perioperative drainage fluids of hepatobiliary pancreatic surgery.

PONE-D-20-20446R1

Dear Dr. Takano,

We’re pleased to inform you that your manuscript has been judged scientifically suitable for publication and will be formally accepted for publication once it meets all outstanding technical requirements.

Kind regards,

Massimiliano Galdiero, M.D., Ph.D.

Academic Editor

PLOS ONE

Additional Editor Comments (optional):

Reviewers' comments:

Reviewer's Responses to Questions

**Comments to the Author**

1. If the authors have adequately addressed your comments raised in a previous round of review and you feel that this manuscript is now acceptable for publication, you may indicate that here to bypass the “Comments to the Author” section, enter your conflict of interest statement in the “Confidential to Editor” section, and submit your "Accept" recommendation.

Reviewer #1: All comments have been addressed

2. Is the manuscript technically sound, and do the data support the conclusions?

Reviewer #1: Yes

3. Has the statistical analysis been performed appropriately and rigorously? 

Reviewer #1: N/A

4. Have the authors made all data underlying the findings in their manuscript fully available?

Reviewer #1: Yes

5. Is the manuscript presented in an intelligible fashion and written in standard English?

Reviewer #1: Yes

6. Review Comments to the Author

Reviewer #1: The authors addressed my comments and followed my suggestions, delivering a new version of the manuscript which requires no more corrections, from my point of view. I recommend the publication of the manuscript in the current format.

7. PLOS authors have the option to publish the peer review history of their article (what does this mean?). If published, this will include your full peer review and any attached files.

Reviewer #1: No

---

## [Editor Report · Acceptance letter]

14 Jan 2021

PONE-D-20-20446R1 

Usefulness of the MALDI-TOF MS technology with membrane filter protocol for the rapid identification of microorganisms in perioperative drainage fluids of hepatobiliary pancreatic surgery. 

Dear Dr. Takano:

I'm pleased to inform you that your manuscript has been deemed suitable for publication in PLOS ONE. Congratulations! Your manuscript is now with our production department. 

Kind regards, 

on behalf of

Prof. Massimiliano Galdiero 

Academic Editor

PLOS ONE